# Eye Temperature Measured with Infrared Thermography to Assess Stress Responses to Road Transport in Horses

**DOI:** 10.3390/ani14131877

**Published:** 2024-06-26

**Authors:** Francesca Aragona, Maria Rizzo, Francesca Arfuso, Giuseppe Acri, Francesco Fazio, Giuseppe Piccione, Claudia Giannetto

**Affiliations:** 1Department of Veterinary Sciences, University of Messina, Polo Annunziata, 98168 Messina, Italy; fraragona@unime.it (F.A.); farfuso@unime.it (F.A.); ffazio@unime.it (F.F.); gpiccione@unime.it (G.P.); clgiannetto@unime.it (C.G.); 2Department of Biomedical and Dental Sciences and Morphofunctional Imaging, University of Messina, Via Consolare Valeria, 98125 Messina, Italy; giuseppe.acri@unime.it

**Keywords:** homeostasis, horse, road transport, eye temperature, cortisol, rectal temperature

## Abstract

**Simple Summary:**

Horses travel frequently during their life activities. Body temperature monitoring is a valuable resource for assessing welfare, physiological state and stress response in mammals, and the eye region offers an ideal location. The current study aimed to address whether infrared measurements of eye temperature may reflect cortisol release in show jumping horses subjected to two different road transport distances. The 100 km journey caused a significant increase in ET, suggesting that the animals did not easily adapt to the new situation in 1 h. The maintenance of the studied parameters was observed during the 300 km journey, reflecting the animals’ adaptation to long-distance transport. This study highlighted the usefulness of IRT as an immediate and non-invasive physiological tool to assess the homeostatic adaptation in athletic horses using an innovative area of interest which allows practical and fast strategies for monitoring the physiological state of the animal during daily activities such as road transport.

**Abstract:**

The aim of the present study was to investigate eye temperature modifications after road transport in athletic horses habituated to travel. Eight adult Italian saddle horses traveled 100 km and, two weeks later, 300 km. Eye temperature (ET), rectal temperature (RT) and serum cortisol concentration were assessed before (T1), after (T2) and 60 min (T3) after the road transport. ET was evaluated with infrared thermography (IRT) in three regions of interest: EL1 (medial canthus), EL2 (central cornea) and EL3 (lateral canthus). Two-way analysis of variance (ANOVA) for repeated measures showed statistically higher values at T2 and T3 for EL1 (*p* < 0.01), EL2 (*p* < 0.01) and EL3 (*p* < 0.01) following the 100 km journey. RT (*p* < 0.01) showed higher values at T2 and T3 after the 100 km journey and higher values at T2 (*p* < 0.01) following the 300 km journey. ET values were positively correlated with RT at T1, T2 and T3 following the 100 km journey and at T2 following the 300 km journey and positively correlated with serum cortisol concentration at T1, T2 and T3 following the 100 km journey and at T2 and T3 following the 300 km journey. Eye temperature monitoring with IRT allows quick and practical strategies to monitor an animal’s physiological state and welfare during daily activities.

## 1. Introduction

Transport activity is an integral aspect of horse management. Horses are transported more frequently than any other type of livestock [1]. They are transported mostly for competitions, breeding, pleasure, sale or slaughter [2]. In the past, horses were transported by train and ship [3,4,5] and also by air [6]. Today, horses are moved mainly by roads with trucks, vans or trailers generally equipped with a rear and/or side ramp for safe loading and unloading [5]. Transport is a daily experience that may influence animals’ homeostasis and welfare. Many factors should be considered during transport practice such as handling, separation from the familiar physical and social environment, loading, vibration, changes in environmental parameters, food and water deprivation, confinement and unloading [7]. All these stimuli trigger responses from the hypothalamic–pituitary–adrenal (HPA) axis, leading to behavioral and physiological changes in the horse such as the secretion of hormones, including via cortisol and hyperthermia mechanisms, which consist of an increase in core body temperature with consequent changes in surface temperature, in order to restore homeostasis [8]. Nevertheless, horses associate transport with their experience before and after transport. Therefore, it is possible that horses transported for competitions and recreational purposes are less adversely affected by transport [9].

The reliable measurement of physiological variations in athletic horses is important both for animal welfare and optimal sport performance [10]. Optimal physiological status evaluation using various invasive techniques, such as blood sampling and cortisol measurement, thermal microchips, tympanic thermometers and rumen boluses, has been well documented in mammals [11,12,13,14]. Research interest has expanded to the use of minimally invasive methodologies to evaluate physiological changes in domestic animals, such as infrared thermography (IRT), which is a non-contact temperature measurement method offering several advantages over other temperature measurement methods used in veterinary medicine [15,16,17,18].

IRT measurement considerably reduces the risk of spreading infection since touching the subject is unnecessary. In animals, this is advantageous since handling and restraint increase stress, causing an effect on core and surface temperatures [19,20,21,22]. It has been widely used in equine veterinary medicine as an auxiliary complementary diagnostic tool for the assessment of musculoskeletal disorders [23], inflammatory conditions and vascular and neurological disorders or for the detection of various limb disorders [13]. Recently, IRT has been used as a reliable method to assess physiological stress in working [24], companion [25] and production animals [26]. IRT can detect the naturally emitted infrared radiation from an object, which is a measure of its surface temperature (ST), providing a representation of it as a heat map [27,28]. This technique is based on using the change in peripheral blood flow to assess variations in the animal’s surface temperature [29]. Changes in core body temperature and skin temperature can be used as an indicator of thermal comfort and welfare [30]. Body temperature is regulated by the thermoregulatory center in the hypothalamus, and it is the result of cellular metabolism, the consequent production of heat and the control of heat dissipation through the body [29]. IRT methods have been attempted across different anatomical areas in different species [19,31,32,33]. It has been argued that the magnitude and direction of temperature changes induced by external stimuli depend on the site of temperature measurement, so the choice of area of interest becomes important [34].

Eye temperature (ET) has been found to be the most consistent region of interest with respect to changes in body temperature compared to other body areas [24]. Measuring eye temperature is quick, relatively easy and accurate as this area is not affected by the presence of hairs compared to other areas [13,35,36]. Eye temperature reflects changes in body temperature in response to events and stimuli. Such stimuli activate the sympathetic nervous system and adrenal gland for the immediate release of catecholamines into the circulating blood stream, promptly raising body temperature [11]. This area is rich in capillary beds, so the measurement of eye temperature allows for the assessment of local changes in blood flow related to the sympathetic nervous system [35,37,38,39]. This association is attributed to the proximity of the orbit to the brain and the rich blood supply it receives. Cortisol is an essential hormone considered an indicator of compromised animal welfare [15]. It is secreted in an ultradian and circadian manner by the fasciculate cells of the adrenal cortex in response to pituitary ACTH stimulation [15,39,40]. Since homeostasis modifications have been shown to cause a marked elevation in cortisol levels, most studies consider cortisol as a biomarker [41,42]. Cortisol release is influenced by several environmental factors, such as time of day, food intake and the psychological state of the animal at the time of sampling [40,43].

In view of such considerations, the current study aimed to investigate whether the eye temperature (ET) assessed by using IRT technique in specific areas (EL1—medial canthus, EL2—central cornea and EL3—lateral canthus) can reflect the body temperature and cortisol release in horses subjected to two road transport distances in order to investigate the usefulness of ET, assessed by IRT, as an innovative tool for investigating the homeostatic changes due to road transport in horses. To achieve this, the correlation between ET, rectal temperature (RT) and cortisol concentration was investigated in adult athletic horses before and after transport.

## 2. Materials and Methods

The study was carried out on 8 clinically healthy Italian saddle horses (4 geldings and 4 non-pregnant mares) aged between 8 and 12 years old with a body weight of 450 ± 50 kg. All horses lived at the same horse training center in Messina, Sicily, Italy (latitude 38°10′35″ N; longitude 13°18′14″ E). They belonged to the same herd and were managed equally during and prior to the experiment. Horses were housed in individual boxes (3.5 × 3.5 m) under a natural photoperiod (sunrise at 05:00; sunset at 21:00) and fed three times a day (06:00, 12:00 and 19:00) with good-quality hay and a mixed cereal concentrate, whereas water was available ad libitum. Prior to the start of both journeys, all horses were considered healthy on veterinary evaluation including routine hematology and biochemistry analyses and were judged fit for transportation. Animal husbandry and data collection were conducted in accordance with European legislation regarding the protection of animals used for scientific purposes recommended by the Guide for the Care and Use of Laboratory Animals and European Directive 2010/63/EU, as recognized and adopted by the Italian law (DL 2014/26). Moreover, each animal was provided for inclusion in the study free of charge by the owner with written informed consent.

### 2.1. Study Design

The animals were subjected to two different road transport journeys to show jumping competitions. All horses traveled in the same truck, tethered with a 50 cm rope on each side of the halter. They were housed in an individual tie stall (length 2.0 m; width 0.85 m) with a total space of 1.7 m^2^. All horses had previous experience of transport. The first journey covered a distance of 100 km with a duration of 1 h and 15 min and an average speed of 80 km/h. The second journey covered a distance of 300 km with a duration of 4 h with an average speed of 75 km/h. Both journeys included a different combination of road surfaces ranging from small country lanes through secondary roads to motorways and started at the same time of day (15:00). The 300 km journey took place two weeks after the 100 km journey, heading for a casually organized sporting event in which the subjects were participating. Moreover, in the two-week period between journeys, no other transport activities were carried out for the subjects. The driver was always the same. No stops were made during transport. The ambient temperature and relative humidity were recorded inside the truck on both journeys using a digital data logger (TH-2500, Gemini Tinytag, Dundee, Scotland) in order to characterize the microclimate of the transport conditions. At the end of both journeys, each horse underwent a clinical examination and did not show any signs of discomfort or the particular degree of sweating that would indicate a dehydration state. The adult horses in question were used to traveling relatively short distances every month for competition/pleasure; likewise, they were accustomed to having routine blood samples taken and did not show fear or reluctance to be approached. The ambient temperature and relative humidity were recorded inside the truck on both journeys using a digital data logger (TH-2500, Gemini Tinytag, Dundee, Scotland) in order to characterize the microclimate of the transport conditions, as shown in Table 1, and the temperature–humidity index (THI) was calculated.

### 2.2. Sampling Collection

A total of 96 thermal images of the left and right eye regions, rectal temperature and blood samples were taken before (T1), after (T2) and 60 min (T3) following the transport. Each measurement was performed at the same time by the same operators after the horses were unloaded and housed in individual boxes. Thermal imaging of the eyes was performed, and an FLIR T440 camera (FLIR Systems, Wilsonville, OR, USA) with a focal plane sensor array size of 320 × 240 pixels was used. To minimize the impact of external stimuli, IRT images were collected in enclosed spaces, free from wind and sunlight exposure. Although the horses were familiar with human interactions, they were curious about the camera; however, they did not resist. The thermal infrared camera was always positioned by the same operator at 40.0 ± 0.1 cm distance, with an angle 90°, from each analyzed area, and the camera was handheld to manage possible muscle movements. Focus, field of view and boundary conditions were automated. Eye temperature (ET) was recorded from the left and right eye perpendicular to the horse’s eye, and an emissivity of ε = 0.98 was set, based on the generally accepted emissivity in a mammal’s eye [44]. The camera acquired images at 60 Hz (60 frames per seconds) with a thermal resolution equal to 0.1 °C and ±2% accuracy of the reading (FLIR user’s manual—Flir T4xx series). The frame rate is the rate at which the infrared detector creates images. Several images were taken per animal and per eye region, with the sharpest, focused and correctly sized images being selected for the analysis following the same selection criteria in order to standardize the samples and avoid possible bias. The stored images were made manually by the operator. All cameras of the same kind have the same infrared properties, irrespective of the frame rate. The thermal time constant is a measure of how quickly the infrared detector itself reacts to fast changes in the incoming radiation. All the acquired images were saved in TIFF format and then converted in lossless JPEG format for later post-processing. The analysis of the raw data was performed by using the FLIR tools (Flir Tools Software). On every single image of the obtained series, the interested region (ROI) was manually drawn and selected. The reproducibility was ensured by using only the FLIR tool’s software, and the ROIs were drawn by a single operator. In particular, the eye was divided into three main ROIs, corresponding to different anatomic structures: EL1 (medial canthus), EL2 (central cornea) and EL3 (lateral canthus). The three eye regions were chosen according to previous studies carried out on horses [10,11,12,13,14,15,16,17,18,19,20,21,22,23,24,25,26,27,28,29,30,31,32,33,34,35,36,37,38,39,40,41,42,43,44,45]. The maximum temperature (T_max_), average temperature (T_avg_) and minimum temperature (T_min_) were automatically extracted from each selected ROI (EL1, EL2 and EL3). Only images which were in focus were used since images which do not meet these requirements can affect the accuracy of the results.

Rectal temperature (RT) was measured using a digital thermometer inserted 15 cm into the rectum (model HI92704, Hanna Instruments, Bedfordshire, UK). The digital thermometer had a 10 s response time and a resolution of 0.1 °C. From each animal, 48 blood samples were collected by jugular venipuncture into vacutainer tubes with clot activator (Terumo Corporation, Tokyo, Japan). Blood samples were centrifuged at 3000× *g* for 10 min. The obtained sera were analyzed to assess the concentration of cortisol using a previously validated enzyme-linked immunosorbent assay (ELISA) kit specific for equine species (cortisol Horse ELISA kit, Abnova, Walnut, CA, USA; KA 2298) by means of a micro-well plate reader (Sirio, SEAC, Florence, Italy) [46]. The sensitivity of the cortisol ELISA kit was 0.5 ng/mL. All calibrators and samples were run in duplicate, and samples exhibited parallel displacement to the standard curve for both ELISA analyses.

### 2.3. Statistical Analysis

All data were analyzed using the statistical software program GraphPad Prism v. 9.5.1 (GraphPad Software Ltd., Solana Beach, CA, USA), and results were considered statistically significant at *p* < 0.05. All data were normally distributed according to Kolmogorov–Smirnov test (*p* > 0.05). A paired Student’s *t*-test was applied to verify statistical differences between left and right eyes. Left and right eyes did not show any statistical differences; hence, the mean value of both eyes has been used for the following analysis. A two-way analysis of variance (ANOVA) for repeated measures was applied to the obtained data to assess significant changes in the ET values of EL1_max_, EL1_avg_ and EL1_min_, EL2_max_, EL2_avg_ and EL2_min_ and EL3_max_, EL3_avg_ and EL3_min_, RT and serum cortisol concentration obtained after both journeys (100 km vs. 300 km) and throughout monitoring times (T1, T2, T3). When significant differences were found, Bonferroni’s test was used for post hoc comparison. Pearson’s correlation coefficients were computed to evaluate the relationship of ET (EL1_max_, EL1_avg_ and EL1_min_, EL2_max_, EL2_avg_ and EL2_min_ and EL3_max_, EL3_avg_ and EL3_min_), RT and serum cortisol concentration at T1, T2 and T3 during both journeys.

## 3. Results

The T_max_, T_avg_ and T_min_ obtained from each ROI (EL1, EL2 and EL3) of the left and right eyes of a representative horse are shown in Figure 1. Two-way ANOVA for repeated measures showed a significant effect of the 100 km journey on ET and RT (*p* < 0.001) and a significant effect of the 300 km journey on RT (*p* < 0.01). In particular, ET increased compared to the baseline during the 100 km journey and remained higher for an hour after arrival. Bonferroni post hoc comparison showed significant higher values of ET and RT at T2 and T3 following the 100 km journey. Statistically higher values of RT were observed at T2 following the 300 km journey. No significant differences due to transport were observed in cortisol concentration, as shown in Table 2. ET values recorded at each ROI (EL1_max_, EL1_avg_ and EL1_min_, EL2_max_, EL2_avg_ and EL2_min_ and EL3_max_, EL3_avg_ and EL3_min_) were positively correlated with RT at T1, T2 and T3 following the 100 km journey and at T2 following the 300 km journey and positively correlated with serum cortisol concentration at T1, T2 and T3 following the 100 km journey and at T2 and T3 following the 300 km journey, as shown in Figure 2.

## 4. Discussion

Knowledge of the functional adaptation of mammals to management practices is critical for monitoring their health and welfare conditions. The ability of an animal to produce an appropriate response to a stimulus that poses a threat to homeostasis is critical for survival. During transport, horses are exposed to a variety of stimuli that can more or less significantly pose a threat to their homeostasis. Environmental parameters recorded and the obtained THI were found to be within a ‘no stress’ area during both journeys. These values can be attributed to the species-specific thermoneutral zone, which is found to be between 10 and 24 °C in a horse where stress is minimal and performance optimal [29,47,48]. The present study showed a significant higher value of ET after the 100 km journey in each investigated eye region and higher values of RT after both journeys, indicating a physiological effect of the transport experience. The protocol was standardized during the experimental phase in fat. No statistical difference was observed between T1 data points after either journey.

The lachrymal caruncule area, proximal to the medial canthus (EL1), is a very sensitive area to both pain and stress events, affecting animals partly due to HPA activation [45]. For this reason, it is known to be the optimal anatomical area for the assessment of ET despite the present study obtaining good results from each analyzed eye region. ET has been widely used to evaluate health status, stress indicators and physiological responses in mammals. In particular, it has previously been assessed as a general indicator to evaluate the individual ability to respond to heat stimulus during transport in different species including horses [38,49,50]. In biological terms, acute psychological stress stimulates sympathetic nervous system to increase heart rate and blood flow through vasoconstriction [39]. The homeostatic response against excess heat leads to a subsequent rise in peripheral temperatures as vasodilation dissipates excess heat [51]. The heat produced by muscles during the first hour of transport is enough to raise the body temperature. In particular, the significant rise in ET obtained after the 100 km journey confirms what was previously observed in other studies on horses [35,43], sheep [8,52] and cattle [53]. Similarly, ET and RT tend to be higher as a result of other stressful events such as exercise in athletic horses [11,50].

Cortisol concentrations did not vary significantly after either journey. Nevertheless, cortisol response magnitude is currently a good standard indicator for stress assessment [41]. In this regard, HPA activation is one of the most important neuroendocrine responses that occurs, promoting the restoration of the body’s homeostasis. Many studies showed cortisol rising following transport [5,42,54,55,56]. These results depend on the habit of the horses, their age and experience, the duration of transport, the environmental conditions and transport management itself. In fact, high cortisol values after transport have been shown in foals during their first experiences over both short and long distances [54,57], in stallions [2] and in racehorses during both long and short journeys [42]. The lack of previous transport experience in subjects used for meat production as opposed to athletic horses used to being transported for competitions presumably contributed to the significant increase in blood cortisol levels detected after transport and slaughter [56]. This may explain the present findings, indicating that cortisol concentration was maintained during the various transport phases. Elevation of cortisol is due to external situations that trigger physiological responses by the body to restore homeostasis. Previous studies have shown that, in some organisms, a minimal initial concentration of cortisol after a stress situation may be sufficient to trigger the desired physiological response to maintain or restore homeostasis [36]. A significant change in ET was only observed during short-term transport. Previous studies reported that animals undergoing a 100 km journey were more stressed than those undergoing a 300 km journey [5,58]. Thus, at the beginning of the journeys, animals were subjected to a form of agitation and anxiety and failed to adapt within 1 h of transport. Previous authors have argued that horses need around 5 h to adapt to a transportation experience, and the first phase is the more critical [58]. Therefore, the adaptation period to a journey is longer than 1 h, which increases the importance of animal management during short journeys [2]. Good management conditions during transport make it possible to maintain an acceptable level of welfare, e.g., choosing an optimal time of day for traveling in accordance with the daily fluctuations of physiological parameters such as body temperature and always encouraging proper ventilation. It would also be good practice to keep horses as close as possible to other horses in the same herd [47,48]. The present results show a positive correlation between ET, RT and cortisol concentration over distances of 100 km. This indicates that these parameters follow the same pattern during transport; therefore, it can be assumed that ocular temperature may be an index of the homeostatic response during road transport activity in athletic horses. Ocular temperature could represent an even more immediate indicator than cortisol, which requires at least 15 min to be sampled [59]. T1 rectal and eye temperatures were correlated for the 100 km transport distance but not for the 300 km transport distance. These differences were not clear; in fact, further studies are suggested to evaluate possible seasonal or circannual effects on ocular temperatures. The use of IRT is a more practical and faster method during activities such as transport, as well as during exercise, where the rapidity with which IRT can be used is certainly optimal compared to RT. In addition, the IRT method provides us with greater utility during those activities when the horse is tied in the truck or is in the field working as it is not easy in the box to monitor rectal temperature. Likewise, this practice can be useful with neurotic, young or traumatized horses that are difficult to approach, and this method is optimal for the safety of the animal as well as the handler. Previous studies demonstrated that both internal and surface body temperature can reveal how animals cope with surrounding environmental changes; thus, it can be hypothesized that eye temperature is a good way to detect hyperthermic changes during transport.

## 5. Conclusions

The present study aimed to identify the usefulness of IRT as an immediate and non-invasive physiological tool to assess the homeostatic modification occurring in athletic horses during road transport using an innovative region of interest, the ocular region. The positive correlation between ocular temperature, rectal temperature and cortisol concentration during both journeys suggests that IRT can provide an objective, rapid and non-invasive assessment during and after transport procedures. In particular, the present results indicate significant modifications in homeostatic responses following the short journey. ET measured by means of IRT allowed an accurate assessment of the state of welfare without the need for invasive procedures such as blood tests. Eye temperature accurately interprets the behavioral and physiological responses of domestic animals during daily activities, such as road transport for competition horses. It seems likely that the adult horses, which were used to traveling for pleasure/competitions on a monthly basis, were not subjected to transport stress but to a normal management routine.

## Figures and Tables

**Figure 1 animals-14-01877-f001:**
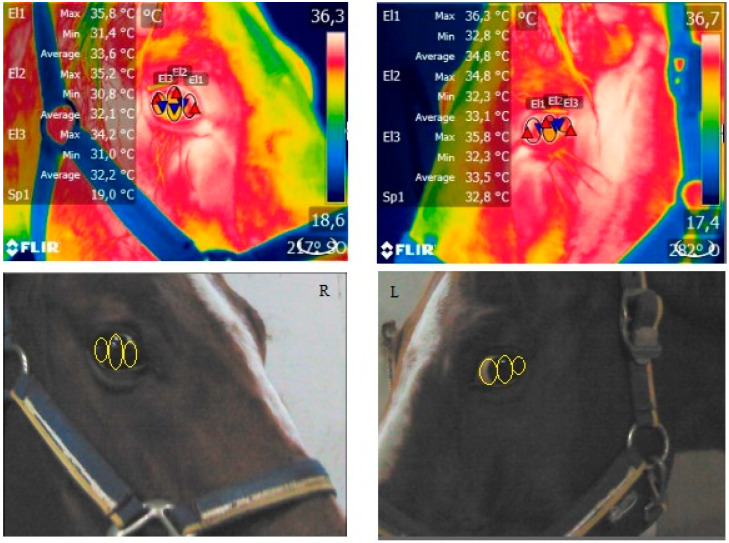
Representative left and right eye with the selected ROIs (EL1 (medial canthus), EL2 (central cornea) and EL3 (lateral canthus)) and the obtained temperatures EL1_max_, EL1_avg_ and EL1_min_, EL2_max_, EL2_avg_ and EL2_min_ and EL3_max_, EL3_avg_ and EL3_min_ in a horse.

**Figure 2 animals-14-01877-f002:**
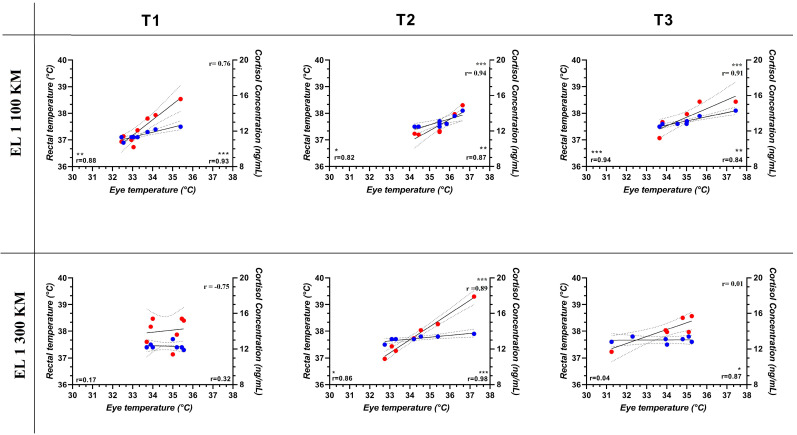
Coefficients of correlation (*r*) among the values of eye temperature (EL1_avg_ chosen as a representative area) [45], rectal temperature (RT) and cortisol concentration, expressed in their conventional unit of measurement, in horses before (T1), after (T2) and 60 min (T3) following journeys of 100 and 300 km. Blue dots represent RT, and red dots represent cortisol concentration. Bottom right: the *r* coefficient refers to the correlation between ET and RT; bottom right refers to the correlation between ET and cortisol concentration. At the top right-hand side, the value of *r* refers to the correlation between RT and cortisol concentration. Symbol * corresponds to *p* < 0.01. Symbol ** corresponds to *p* < 0.001. Symbol *** corresponds to *p* < 0.0001.

**Table 1 animals-14-01877-t001:** Environmental conditions recorded inside the truck, expressed in their conventional unit, during the two different journeys (100 km and 300 km).

Environmental Factors	Experimental Conditions	Max.	Mean	Min.
Ambient temperature (°C)	100 km	24.75	21	18.13
	300 km	27.6	22	20.7
Relative humidity (%)	100 km	79.30	67.50	51.64
	300 km	80.32	63.80	51.96
Temperature–humidity index (THI)	100 km	66.21		
	300 km	69.19		

**Table 2 animals-14-01877-t002:** Mean ± Standard Deviation (SD) of ET (EL1_max_, EL1_avg_ and EL1_min_, EL2_max_, EL2_avg_ and EL2_min_ and EL3_max_, EL3_avg_ and EL3_min_), RT and cortisol concentration recorded before (T1), immediately after (T2) and 1 h after (T3) the 100 km and 300 km journeys. Significant differences are expressed by *, corresponding to *p* < 0.01; ** *p* < 0.001.

Parameters			100 km			300 km		
			T1	T2	T3	T1	T2	T3
eye temperature (°C)	EL1	max	35.3 ± 1	36.8 ± 0.8 *	36.7 ± 0.7 *	36.7 ± 0.60	36.3 ± 1.42	35.9 ± 1
		avg	33.4 ± 1	35.5 ± 0.8 **	35 ± 1.2 *	34.7 ± 0.79	34.4 ± 1.56	33.8 ± 1.51
		min	31.7 ± 1.3	33.4 ± 0.7 *	33.2 ± 1.1 *	33.1 ± 0.60	32.7 ± 1.38	31.6 ± 1.79
	EL2	max	33.7 ± 1.3	34.9 ± 1.1 *	35.1 ± 0.9 *	35 ± 1.16	35.1 ± 1.46	34.3 ± 1.29
		avg	32.3 ± 1.2	33.6 ± 0.9 *	33.9 ± 1.2 **	33.4 ± 1.04	33.4 ± 1.21	32.6 ± 1.29
		min	31.3 ± 1.8	33 ± 0.9 *	32.9 ± 1.3 *	32.4 ± 1.21	32.3 ± 1.37	31.6 ± 1.46
	EL3	max	34.5 ± 1	35.7 ± 1 *	35.8 ± 1 *	35.4 ± 0.98	35.6 ± 1.25	35.2 ± 0.65
		avg	33.1 ± 0.9	34.6 ± 1.1 **	33.5 ± 1.3 *	34 ± 1.35	34.3 ± 1	33.7 ± 0.88
		min	31.7 ± 1	33.4 ± 1.1 **	34.5 ± 1.1 *	32.8 ± 1.44	33.1 ± 1.20	32.3 ± 1.12
rectal temperature (°C)			37.5 ± 0.1	37.7 ± 0.2 *	37.7 ± 0.2 *	37.4 ± 0.13	37.7 ± 0.13 *	37.7 ± 0.11
cortisol (ng/mL)			12.3 ± 1.8	12.7 ± 1.2	13.6 ± 1.4	13.8 ± 1.53	13.6 ± 2.33	14 ± 1.3

## Data Availability

The raw data supporting the conclusions of this article will be made available by the authors on request.

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
