# Peer review of "Eye Temperature Measured with Infrared Thermography to Assess Stress Responses to Road Transport in Horses"

_animals, 2024, doi:10.3390/ani14131877_

Round 1
Reviewer 1 Report
Comments and Suggestions for Authors
Eye temperature in horses
I accept the manuscript after major revision
General comments
The manuscript describes an interesting and well performed study that could be useful for the evaluation of handling procedures in domestic animals.
Some of the text is too detailed. For instance, blood sampling etc. (line 187-190) could be shortened. The many ref.s to cortisol measurements should be shortened (line 58-61), as well as the many ref.s in line 81. Other places are mentioned below. In addition, generally the text needs correction by an English speaking person.
Specific comments
Title: I recommend the following change: Eye temperature measured with infrared thermography to assess stress responses to road transport in horses
Line 25 Change ‘used’ to ‘habituated’
Line 25-26 Delete for: …travelled 100 km and, two weeks later, 300 km.
Line 48-49 Delete one confinement
Line 54-57 How do the authors know that competitions and recreational purposes are positive experiences for horses? Maybe they are only positive for the riders. Change sentence to: Therefore, it is possible that horses transported for competitions and recreational purposes are less adversely affected by transport
Line 59 Change have to has
Line 60 Change ‘such as blood sampling, cortisol measurement’ to ‘such as blood sampling and cortisol measurement’
Line 71 Don’t use ref when you use number elsewhere. Change to 13
Line 88 Change ‘as’ to ‘since’
Line 90-93 An awkward sentence. Delete it, it’s not necessary.
Line 100 Change ‘investigated’ to ‘investigate’
Line 124 Change of to with
Line 125 Delete length
Line 126 Delete length
Line 129 Change ‘started at the same time of day’ to ‘and started at the same time of day’
Line 132 Change in to on
Line 134 Change ‘showed’ to ‘shown’
Line 135-149 Delete table 1 and text ‘The temperature- humidity index… relative humidity- % [44- 45].’ Since the values are within normal range we don’t need the details.
Line 153 Change ‘were’ to ‘was’ Change ‘since’ to ‘after’
Line 155 Change ‘with’ to ‘and’
Line 159 Add the: The thermal infrared camera
Line 162 Change were to was
Line 164 Change ‘in mammals body’ to ‘in a mammal’s eye’ Ref. not in the list. Use number.
Line 168 Change ‘body’ to ‘eye’
Line 180 Use number
Line 183 Change ‘as’ to ‘since’
Line 200 Change ‘resulted’ to ‘were’
Line 201 Change sentence to: A paired student's t-test
Line 202-203 The sentence is awkward
Line 203 Add an a: A two-way analysis
Line 206 Change sentence to: and on serum cortisol concentrations obtained
Line 220-221 If there are no significant changes in cortisol concentrations it does not have to be shown in the figures. It is enough to mention it in the text.
Line 229 Add a: …in a horse
Line238 Change sentence …’ immediately after (T2) and after1 hour (T3) after the road transport’ to …’immediately after (T2) and 1 hour after (T3) the road transport’
Line 243 Same problem as above
Line 274 Change ‘in each eye investigated region’ to’ in each investigated eye region’
Line 279 Change ‘handle’ to ‘evaluate’
Line 280 Add s on indicator: indicators
Line 282-290 This section is basic knowledge that does not have to be repeated in the article. In addition, the meaning of the text is not very clear.
Line 291 Add s to concentration
Line 297 How does destination affect cortisol increase? Don’t the authors mean which activity the horses participate in at the destination may affect cortisol secretion?
Line 301 What is meant with ‘and their temper’? If the authors mean something like temperament, I think it will be better just to delete the three words to keep the article short.
Line 305 Change ‘reluctance to approach’ to ‘reluctance to be approached’
Line 314 Change ‘fails’ to ‘failed’
Line 318 Change ‘A good management’ to ‘Good management’
Line 320 Change ‘showed’ to ‘show’
Line 324 Delete ‘be going to’
Line 333 Add the: the ocular region
Line 335 Change ‘suggested’ to ‘suggests’
Line 337 Change ‘indicated’ to ‘indicate’ Line 340 Add s to interpret
Line 341 Change ‘activities in particular for competition’ to ‘activities, such as road transport for competition horses’
Line 342 Change ‘It appeared evident that’ to ‘It seems likely that’
Comments on the Quality of English Language
see above
Author Response
Thanks for the detailed reviews, please see the attached file for comments and responses to reviewers

Reviewer 2 Report
Comments and Suggestions for Authors
Please see attached file

Comments on the Quality of English LanguageThe manuscript has numerous, but relatively minor, grammatical and typographical errors (and some line spacing errors). In some places word choice or grammatical structure changes the meaning. Most of these can be corrected with close editing.
A few illustrative examples are listed below.
Line 22: ‘region’ in English generally refers to geographical location. A more general term might better suit the intention, such as area or topic.
Line 56. Adverse is used as an adjective in this text thus should be adversely (this type of error appears several times).
Line 94: The passage is oddly phrased. A possible alternative might be: "cortisol is an essential hormone considered an indicator of compromised animal welfare"?
Line 153: Example of one plurality grammatical error: “Each measurement were was performed’ at the same time…’
Author Response

(The authors gave the same response as above.)
